# Alterations in the Level of Ergosterol in *Candida albicans*’ Plasma Membrane Correspond with Changes in Virulence and Result in Triggering Diversed Inflammatory Response

**DOI:** 10.3390/ijms24043966

**Published:** 2023-02-16

**Authors:** Daria Derkacz, Anna Krasowska

**Affiliations:** Faculty of Biotechnology, University of Wroclaw, 50-383 Wroclaw, Poland

**Keywords:** *Candida albicans*, Erg11p, biofilm, filamentation, adhesins, inflammatory response

## Abstract

Opportunistic pathogen *Candida albicans* possesses multiple virulence factors which enable colonization and infection of host tissues. *Candida*-related infections frequently occur in immunocompromised patients, which is related to an insufficient inflammatory response. Furthermore, immunosuppression and multidrug resistance of *C. albicans* clinical isolates make the treatment of candidiasis a challenge for modern medicine. The most common resistance mechanism of *C. albicans* to antifungals includes point mutations in the *ERG11* gene, which encodes target protein for azoles. We investigated whether the mutations or deletion of the *ERG11* gene influence the pathogen-host interactions. We prove that both *C. albicans erg11∆/∆* and *ERG11K143R/K143R* exhibit increased cell surface hydrophobicity. Additionally, *C. albicans* KS058 has an impaired ability of biofilm and hyphae formation. Analysis of the inflammatory response of human dermal fibroblasts and vaginal epithelial cell lines revealed that altered morphology of *C. albicans erg11∆/∆* results in a significantly weaker immune response. *C. albicans ERG11K143R/K143R* triggered stronger production of pro-inflammatory response. Analysis of genes encoding adhesins confirmed differences in the expression pattern of key adhesins for both *erg11∆/∆* and *ERG11K143R/K143R* strains. Obtained data indicate that alterations in Erg11p consequence in resistance to azoles and affect the key virulence factors and inflammatory response of host cells.

## 1. Introduction

The polymorphous fungus *Candida albicans* constitutes part of the natural microbiome of healthy individuals [1]. It can be found in several parts of the body, e.g., the intestine, skin, mouth, or vagina, without causing any diseases. Nevertheless, prolonged usage of antibiotics, chemotherapy or immunosuppression predestines the development of *Candida*-related infections [2,3]. Among different species of *Candida*, *C. albicans* remains the main cause of fungal infections. Candidemia (bloodstream infection) is responsible for 10% of nosocomial infections, and the mortality rate ranges from 30 to 60% [4,5]. There are many difficulties during the treatment of candidemia and candidiasis (mucosal infections). A limited number of available therapies and potential drug targets or the emergence of multi-drug resistant *Candida* sp. led to a high mortality rate among patients suffering from fungal infections.

Successful colonization and infection of host tissues are provided by multiple virulence factors of *C. albicans*. One of them is the ability of this fungus to morphological transition from yeast (blastoconidia) to hyphae [6]. Filaments enable active invasion and penetration of host tissues. Additionally, recent studies also show that *C. albicans* produces a hyphae-specific toxin (candidalysin), which is responsible for damage of epithelial cells and triggering the innate immune response, which results in cytokine production [7,8]. Apart from morphological transitions, *C. albicans* is able to form a biofilm that provides resistance to antifungals. This specialized structure is composed of *C. albicans* cells embedded in extracellular polysaccharide substance (EPS) attached to host tissues or abiotic surfaces [9]. The adhesion to different surfaces is also facilitated by *C. albicans* cell wall (CW) proteins (adhesins). The best-known adhesin family is an agglutinin-like sequence (Als), e.g., Als1p or Als5p [10]. There are also hyphae-specific adhesins like hyphal wall protein 1 (Hwp1p), in which the N-terminal end constitutes a substrate for human transglutaminases providing the covalent bonding to host tissues [11]. Despite the fact that adhesion is the first and critical moment of *C. albicans* infection, there are several different factors that enable the successful colonization of host tissues. The quorum sensing, creation of opaque cells and changes in cell surface hydrophobicity (CSH) also affect the invasive abilities of *C. albicans* [12,13].

During infection *C. albicans* CW is the first barrier to interfere with the host cells’ surface. According to this, CW plays a major role in initial colonization (e.g., adhesins) but also stimulates the innate immune system to fight the infection [14]. *C. albicans* CW is composed of a thick layer of mannans (outer CW), β-glucan and chitin (inner CW). Those components are considered pathogen-associated molecular patterns (PAMPs) [15]. Nevertheless, a major role in the stimulation of inflammatory responses is played by mannans and β-glucan, which are recognized by host pathogen-recognition receptors (PRRs) [16]. Interestingly, in some conditions (e.g., acidic environment, presence of lactate, antibiotic treatment or gene manipulation), *C. albicans* remodels its CW to evade the host’s immune system [17]. Major *C. albicans* PAMP β-glucan can be both exposed or masked in response to environmental changes [18,19]. Nevertheless, recognition of PAMPs by host PRRs leads to the activation of specific signaling pathways, which results in the production of proinflammatory cytokines [20]. Since *C. albicans* infection frequently occurs in the mucosal epithelium (oral or vagina epithelium), these cells produce inflammatory interleukins (ILs), e.g., IL-1α, IL-6, IL-10 or IL-8 [21,22]. IL-8 is a strong chemoattractant to phagocytes which constitutes a significant line of defense against *C. albicans* infection [23]. Considering that fungal cells are too large to be phagocytized (e.g., hyphae), the creation of neutrophil extracellular traps (NETs) is an effective way to combat *C. albicans* infections [24]. Attracted to the site of infection, neutrophils release DNA with histones and antimicrobial peptides, which immobilize and kill pathogens [25].

There are many different strategies to eliminate fungal infections, but the crucial part is the recognition of PAMPs and the production of proinflammatory cytokines. On the other hand, *C. albicans* possess multiple virulence factors and strategies to evade the immune response. Additionally, considering the growing multidrug resistance of *C. albicans* clinical isolates, there is an uprising problem with successful treatment. The most commonly employed antifungals, like azoles (e.g., fluconazole), target the Erg11p (encoded by the *ERG11* gene), resulting in blocking the ergosterol biosynthesis pathway and growth inhibition of the pathogen. *C. albicans*’ resistance strategies to azoles include point mutations in the *ERG11* gene or alterations to *ERG11* expression [26,27].

We previously reported that both point mutation (*K143R*) in the *ERG11* gene (in *C. albicans* 10C1B1I1 strain) or deletion of this gene (in *C. albicans* KS058 strain) corresponds with changes in the plasma membrane (PM) and CW composition of *C. albicans* mutants [28]. CW remodeling can impact both virulence of *C. albicans* and the immune response. Thus, we decided to investigate whether introduced mutations affect inflammatory response.

Here, we investigated the cell surface hydrophobicity (CSH) of *C. albicans* KS058 (*erg11∆*/*∆*) and 10C1B1I1 (*ERG11^K143R^*^/*K143R*^). Additionally, we analyzed the filamentation and biofilm formation of *C. albicans* strains in order to determine the impact of the mutations on the virulence of the pathogen. Since *C. albicans* lives on skin and mucosal epithelium (e.g., vagina), we investigated the inflammatory response of normal human dermal fibroblasts (NHDF) and normal vaginal epithelial (VK2 E6/E7) cell lines to *C. albicans* mutants’ infection. Research revealed that *C. albicans* KS058 and 10C1B1I1 strains lead to altered inflammatory response and different cytokines expression patterns in the NHDF and VK2 E6/E7 cell lines. Additionally, we determined the expression of adhesins for KS058 and 10C1B1I1 strains cultured in YPD medium and after co-culture with NHDF and VK2 E6/E7 cell lines. Obtained data suggest that alterations in ergosterol levels in PM lead to significant changes in the virulence of *C. albicans* and in triggering inflammatory responses during infection. Therefore, a continuation of the study will impact the understanding of the consequences of *C. albicans*’ resistance mechanisms on host-pathogen interactions.

## 2. Results

### 2.1. C. albicans Mutants with Altered Ergosterol Content Display Elevated Cell Surface Hydrophobicity (CSH) and Changed Ability of Biofilm Formation

The aim of this study was to investigate whether changes in the CW composition of *C. albicans* KS058 (*erg11∆*/*∆*) and 10C1B1I1 (*ERG11^K143R^*^/*K143R*^) mutants [28] correlate with altered cell surface hydrophobicity (CSH) and formation of biofilm. The results of CSH and biofilm assay are presented in Figure 1.

For both *C. albicans* KS058 and 10C1B1I1 strains, the CSH parameter was significantly elevated in comparison to the SC5314 (WT) strain (for WT, KS058 and 10C1B1I1, the CSH were 38.92; 72.50 and 77.95%, respectively). For this study, we used the hydrophobic CAF4-2 strain (parental strain: CAF2-1) as an internal control. The obtained data show that, for the *C. albicans* CAF4-2 strain, the surface hydrophobicity is 61.47%. Analysis revealed that *C. albicans* KS058 and 10C1B1I1 strains exhibit significantly elevated cell surface hydrophobicity (72.50 and 77.95%, respectively). Those results indicate the crucial role of ergosterol in maintaining fungal cell physiology.

Furthermore, the lack of ergosterol in *C. albicans* KS058 resulted in impaired biofilm formation compared to the SC5314 (WT) strain (Figure 1B). After 24 h of culture in a RPMI 1640 medium, the *C. albicans* 10C1B1I1 strain exhibited elevated biofilm formation. According to this data, the amino acid substitution *K143R* in the *C. albicans* 10C1B1I1 strain leads to elevated ergosterol content in the PM and CW remodeling and affects the formation of biofilms as one of the key virulence factors.

### 2.2. C. albicans erg11∆/∆ Has an Impaired Ability of Hyphae Formation

Polymorphism of *C. albicans* is an essential aspect of fungal infections. The capacity for morphological transitions from yeast to hyphae enables dissemination, invasion and the damage of host tissues [6]. Thus, we performed a filamentation assay in order to find out whether manipulations in Erg11p (the target for commonly used antifungals, azoles) correlate with significant changes in this phenomenon. Microscopic analysis and measurements of the hyphae length of *C. albicans* WT (SC5314), KS058 and 10C1B1I1 strains after culture in the presence of FBS are presented in Figure 2.

Analysis revealed that *C. albicans* KS058 is not able to form hyphae during 24 h of culture in the presence of FBS. This proves that lack of ergosterol affects the crucial virulence factors of *C. albicans* (Figure 1B and Figure 2) and leads to significant morphological changes in the KS058 strain.

According to Table 1, the percent of hyphae cells for *C. albicans* 10C1B1I1 (*n* = 300) was elevated only after 2 h of culture compared to the WT strain (72.29 and 64.82 %, respectively).

In most of investigated time points (t = 1, 4, 6, 8 and 24 h), the percentage of hyphae in culture was lower or similar for *C. albicans* 10C1B1I1 compared to the WT strain. On the other hand, elevated ergosterol content (as a consequence of amino acid substitution in Erg11p) in *C. albicans* 10C1B1I1 mutant resulted in the formation of longer filaments than in WT strain but only in the early logarithmic phase of growth (t = 2, 4, 6 and 8 h of culture). (Figure 1B).

### 2.3. Alterations in PM and CW of C. albicans Ergosterol Mutants Correlates with Different Patterns of Inflammatory Response and Cytokines Genes Expression of Host Cell Lines

The obtained results indicated that both the depletion and increased levels of ergosterol in *C. albicans* PM correlate with altered virulence potential. We employed analysis of inflammatory response to infection with *C. albicans* mutants. Since *C. albicans* usually live on skin or mucosal epithelia (e.g., mouth or vagina epithelium), we determine the cytokine release of normal human dermal fibroblasts (NHDF) and normal vaginal epithelial (VK2 E6/E7) cell lines (Table 2).

Research revealed that NHDF cell lines do not produce IL-1α either in control conditions or during infection with *C. albicans*. Stimulation of NHDF with pathogen mutants led to the production of IL-6, Il-8, MCP-1 and trace amounts of IL-10 (Table 2). IL-6 production was significantly elevated after co-culture with WT and 10C1B1I1 strains after 8 h of co-culture (20.45 and 24.14 pg/mL, respectively), while culture with KS058 did not stimulate cells for increased production of IL-6 compared to control conditions (2.60 and 1.48 pg/mL, respectively). The IL-6 production after 24 h of co-culture significantly increased for co-culture with all tested strains, while the KS058 strain triggered the poorest IL-6 release in the NHDF cell line. We obtained similar results in the case of chemokine MCP-1; an increased level of MCP-1 was detected for the WT and 10C1B1I1 strains (13.98 and 18.04 pg/mL). *C. albicans* KS058 did not stimulate NHDF cells to produce MCP-1 (6.00 pg/mL). Interestingly, after 24 h of co-culture, we observed significantly elevated levels of MCP-1, both in control conditions (277.38 pg/mL) and after co-culture with *C. albicans* WT, KS058 or 10C1B1I1 strains (476.25; 314.52 or 405.78 pg/mL, respectively). The level of IL-8 was comparable to control conditions for the KS058 strain (0.15 and 0.39 pg/mL, respectively). A higher concentration of IL-8 was observed in the case of NHDF co-cultured with *C. albicans* WT or 10C1B1I1 strains (3.16 or 8.69 pg/mL, respectively). The amount of IL-8 significantly increased after 24 h of co-culture with WT and 10C1B1I1 strains (39.52 and 52.34 pg/mL, respectively). Thus, *C. albicans* 10C1B1I1 strain triggers a stronger inflammatory response in the NHDF cell line after 24 h of culture than the WT strain. On the other hand, *C. albicans* KS058 did not stimulate NHDF to produce higher levels of the tested cytokines.

The analysis of the inflammatory response of the VK2 E6/E7 cell line revealed that co-culture with *C. albicans* WT and 10C1B1I1 strains triggered the significantly higher production of IL-1α, while for *C. albicans* KS058, the level of IL-1α was comparable to the control conditions after 8 h of co-culture (1612.60; 1530.79 and 5.66 pg/mL, respectively). 24 h post-infection, the IL-1α level in co-culture supernatant increased in the case of all tested *C. albicans* WT, KS058 and 10C1B1I1 strains (1713.36; 59.18 and 2087.91 pg/mL, respectively). The VK2 E6/E7 cell line, after co-culture with *C. albicans* mutants, also produced an increased amount of IL-6 and IL-8. We observed that the WT and 10C1B1I1 strains triggered IL-6 production more effectively than the KS058 strain after 8 h of co-culture (13.13, 16.66 and 1.38 pg/mL, respectively). In 24 h of culture, we observed an increase in IL-6 levels in the presence of *C. albicans* KS058 (6.64 pg/mL) and the results obtained for the WT and 10C1B1I1 strains were comparable to those achieved in 8 h of culture. We observed a similar pattern in the case of IL-8; the stronger response of the VK2 E6/E7 cell line to the *C. albicans* WT and 10C1B1I1 strains, and the level of this chemokine was comparable among the investigated times of co-culture (8 and 24 h). Moreover, the *C. albicans* KS058 strain did not stimulate vaginal epithelial cells to increase production of IL-8 after 8 h of culture, but the elongation of the time of co-culture (from 8 to 24 h) with this strain resulted in elevated levels of IL-8 (3.78 and 24.71 pg/mL, respectively). Similarly to NHDF, the VK2 E6/E7 cell line after *C. albicans* infection produced trace amounts of IL-10 in all tested conditions. The level of MCP1 was beyond the value obtained for the control condition in the case of all *C. albicans* strains, while the highest level of this chemokine was noted for *C. albicans* KS058 strain (after 8 and 24 h of co-culture MCP1 level was as follows: 4.90 and 3.31 pg/mL).

Additionally, we investigated the relative expression of genes encoding chosen cytokines (IL-6, IL-8 and IL-1α) in order to confirm obtained results. The experiment was performed using the same cells harvested for analysis of cytokines production (Table 2). The relative expression of genes encoding IL-6, IL-8 and IL-1α are presented in Figure 3.

The research indicates that stimulation of the NHDF cell line with *C. albicans* WT or 10C1B1I1 strains for 8 h results in the increased expression of *IL-6* and *IL-8* genes (Figure 3A). At the same time, the higher expression of the *IL-8* gene was observed for co-culture with *C. albicans* 10C1B1I1 compared to the WT strain. We observed about a 10-fold increase in the expression of *IL-6* and *IL-8* after 24 h of co-culture NHDF with WT and 10C1B1I1 strain (Figure 3B). *C. albicans* KS058 induced the significantly lower expression of genes encoding investigated cytokines in both NHDF and VK2 E6/E7 cell lines. After 8 h of culture, the level of expression of the *IL-6* gene was slightly increased, and higher expression was observed for the WT strain compared to the 10C1B1I1 strain (Figure 3C). Nevertheless, after 24 h of culture, this expression was elevated in VK2 E6/E7 cells stimulated with *C. albicans* WT and 10C1B1I1 strains, and we noted the higher expression of *IL-6* in the case of the 10C1B1I1 strain (Figure 3D). This data also supports the results obtained in the investigation of cytokines production (Table 2). The expression of the *IL-8* gene in VK2 E6/E7 cells after 24 h of stimulation with *C. albicans* 10C1B1I1 was significantly higher than after stimulation with the WT strain (Figure 3D). After 8 h of co-culture with the WT or 10C1B1I1 strains, we observed a similar level of *IL-8* expression (Figure 3C).

### 2.4. C. albicans erg11∆/∆ and ERG11^K143R/K143R^ Strains Display a Different Expression Pattern of Genes Encoding Crucial Adhesins

Considering that adhesins are crucial for the initial steps of the *C. albicans* infection, we investigated the relative expression of genes encoding those proteins (*ALS1*, *ALS5*, *HWP1* and *SAP5*) for the purpose of determining if there are differences in adhesins’ expression profile between *C. albicans* mutants.

Firstly, we determined the gene expression of adhesins after *C. albicans* WT (SC5314), KS058 and 10C1B1I1 strains culture in the YPD medium. The results are presented in Figure 4.

In the case of the *C. albicans* KS058 strain, we observed higher expression of all investigated genes encoding adhesins after 8 h of culture in the YPD medium (Figure 4A). For *C. albicans* KS058, the highest expression was detected for the *HWP1* gene (about a 10-fold increase in relative *HWP1* expression). The *C. albicans* 10C1B1I1 strain exhibited significantly increased expression of *ALS5* and *HWP1* genes, while the *HWP1* expression was six-fold higher than for the WT strain after 8 h of culture.

After 24 h of culture in YPD medium, the expression of *ALS5* and *HWP1* genes decreased compared to 8 h of culture and is similar for all investigated strains, except the two-fold higher expression of *ALS5* for *C. albicans* KS058 compared to the WT strain (Figure 4B). Interestingly, the *ALS1* gene expression after 24 h was significantly higher than after 8 h of culture for KS058 (a 10-fold increase compared to the WT strain).

Additionally, we determined the adhesins gene expression profile in *C. albicans* WT (SC5314), KS058 and 10C1B1I1 strains after co-culture with NHDF and VK2 E6/E7 cell lines. The results are presented in Figure 5.

After 8 h of co-culture with the NHDF cell line, the *C. albicans* 10C1B1I1 strain displayed lowered *ALS1* gene expression compared to the WT strain. The expression of *ALS1* and *ALS5* for *C. albicans* KS058 was elevated but not statistically significant (Figure 5A). There were no significant differences in the expression of genes encoding adhesins among investigated *C. albicans* strains after 24 h of co-culture with the NHDF cell line (Figure 5B). In the case of co-culture with VK2 E6/E7 cell line for 8 h, there was no change in *ALS1* gene expression. On the other hand, the expression of *ALS5* and *HWP1* genes was significantly lower in *C. albicans* KS058 and 10C1B1I1 compared to the WT strain (Figure 5C). An extended time of co-culture (24 h) resulted in an increase in both *ALS1* and *HWP1* gene expression in the *C. albicans* 10C1B1I1 strain (Figure 5D). *C. albicans* KS058 displayed a significant increase in *ALS5* gene expression compared to 8 h of culture.

## 3. Discussion

*Candida*-related infections are responsible for the high mortality rate of hospitalized patients ranging from 30 to 80% [29,30]. Bloodstream infections (candidemia) result in mortality of 30 to 60% of individuals [4]. That problem is related to the multidrug resistance phenomenon among clinical isolates of *Candida* sp. The most common strategies to gain resistance to azoles are mutations in the *ERG11* gene, which encodes the enzyme Erg11p [31]. This protein is responsible for the demethylation of lanosterol in the ergosterol biosynthesis pathway. Mutations in *ERG11* (decreased or increased expression) not only provide resistance to azoles but can also affect the morphology of fungal cells. Therefore, it is relevant to study mechanisms involved in acquiring resistance to antifungals and to determine the impact of the mutations on fungal pathogenicity.

We previously have shown that amino acid substitution K143R in Erg11p results in both increased *ERG11* gene expression and elevated ergosterol content in *C. albicans* PM [28]. More interestingly, K143R substitution (in *C. albicans* 10C1B1I1) and the lack of Erg11p (in *C. albicans* KS058) result in the exposure of chitin and β-glucan on the cell surface [28,32]. Additionally, both of these mutations (depletion and increased expression of the *ERG11* gene) result in increased azole resistance. Taking those data into account, *C. albicans* 10C1B1I1 and KS058 strains could be relevant models in studying the inflammatory response of human cells to infection with azole-resistant *C. albicans* strains.

The purpose of the study was to verify whether K143R substitution in Erg11p and lack of this protein influence the key virulence factors of *C. albicans*, e.g., cell surface hydrophobicity (CSH), filamentation and biofilm formation. The surface of fungal cells is important during the initial infection of host tissues because it is the first one to interfere with human cells. The analysis revealed that, compared to *C. albicans* WT (SC5314), both the KS058 and 10C1B1I1 strains demonstrated significantly higher CSH values, and the CSH obtained for those mutants was even higher than for hydrophobic CAF4-2 strain (Figure 1A). According to these data, the K143R substitution or lack of Erg11p results in a substantial increase in the CSH parameter. We previously reported that both *C. albicans* KS058 and 10C1B1I1 strains display an altered CW structure by exposure to β-glucan and chitin on the cell surface [28]. The CSH has been linked to the character of the CW architecture [33]. Therefore, the alteration in *C. albicans* mutants CW (different composition of CW components and potential alteration in the CW protein architecture) can impact the CSH of investigated strains. Our previous studies confirmed that elevated CSH results in higher activity of the Cdr1p efflux pump leading to increased fluconazole resistance [34]. Additionally, the CW is the most outer compartment of fungal cells, then alterations in the composition of the CW could impact the overall hydrophobicity of the cells. Interestingly, the exposure of surface proteins (e.g., proteases) in *Candida* CW leads to decreased CSH [35]. Moreover, the cells of *Candida* sp., which display high CSH values, are more virulent than hydrophilic cells in mice models, and increased CSH is highly related to disease-causing isolates [36].

One of the key virulence factors of *C. albicans* is its ability to form biofilms both on biotic and abiotic surfaces, which can be critical for patients after transplantations [37]. Here, the biofilm formation assay analysis revealed that *C. albicans* KS058 has an impaired ability of biofilm formation (Figure 1B). This could be related to decreased growth rate due to the depletion of ergosterol from PM of *C. albicans* KS058 [28]. Opposite to the KS058 strain, *C. albicans* 10C1B1I1 demonstrates a slightly increased ability to form biofilms compared to the WT strain. In conclusion, the alterations in ergosterol content impact the biofilm formation and attachment of fungal cells to abiotic surfaces. This could also influence the adhesion of *C. albicans* mutants during the infection of host tissues. The diminished ability to form biofilm is commonly related to the altered expression of adhesin genes (e.g., *ALS1*, *HWP1*), filamentation or CSH [38,39].

The filamentation assay revealed that, for *C. albicans* WT and 10C1B1I1 strains, the formation of filaments initiated after 1 h of culture in YPD media supplemented with 20% FBS (Figure 2A). After 4 h of culture, a major number of *C. albicans* WT cells produced hyphae, while, for the *C. albicans* 10C1B1I1 strain, we observed a mixed culture of blastoconidia and hyphae. Additionally, in the early phase of culture, the median hyphae length was significantly higher for *C. albicans* 10C1B1I1 compared to the WT strain (Figure 2B). On the contrary, *C. albicans* KS058 did not grow as filaments in any of the investigated time points, which indicates that the lack of ergosterol contributes to impaired ability of filamentation. In *Candida* sp., filamentous growth can be influenced by a vast number of factors, e.g., regulation via sterol regulatory-element binding protein (SREBP) or by the production of hyphae-specific adhesins (e.g., Hwp1p) [40,41]. For the first time, to our knowledge, we show the correlation between ergosterol content and filamentous growth. In conclusion, the alterations in the level of ergosterol led to substantial changes in the hydrophobicity of fungal cells, their ability to form biofilms and filamentous growth.

*C. albicans* KS058 and 10C1B1I1 also demonstrate different patterns of expression of genes encoding adhesins. After 8 h of culture in YPD media, we observed increased expression of *ALS1*, *ALS5*, *HWP1* and *SAP5* for *C. albicans* KS058 or *ALS1* and *HWP1* for *C. albicans* 10C1B1I1 (Figure 4A). The significant increase in *HWP1* gene expression is surprising in the case of the *C. albicans* KS058 strain because this strain does not grow as hyphae (Figure 2). This suggests that the KS058 strain upregulates the expression of the adhesins genes as a response to morphological changes caused by ergosterol depletion (e.g., impaired biofilm formation). The expression of *ALS1* and *SAP5* was elevated when KS058 cells were in the stationary phase of growth (Figure 4B). Surprisingly, the expression of *HWP1* for all tested strains decreased after 24 h of culture in YPD media. The adhesins are GPI-anchored proteins of *C. albicans* CW, providing the successful colonization of host tissues [42]. Thus, we show that alterations in CW composition in *C. albicans* mutants impact the expression of genes encoding adhesins.

Taking that into account, we assumed that arising changes would be crucial for triggering the inflammatory response. In order to verify this statement, we performed co-cultures of *C. albicans* mutants with normal human dermal fibroblast (NHDF) and normal vaginal epithelial VK2 E6/E7 cell lines. The chosen cell lines represent the host niches for *Candida* sp. We investigated the expression of adhesins, level of cytokines (IL-1α, IL-6 and IL-10) and chemokines (IL-8 and MCP-1) which are produced by human cells in response to *C. albicans* infection (Table 2).

In the case of *C. albicans* KS058 or 10C1B1I1 co-culture with NHDF, we did not observe significant changes in the expression of *ALS1*, *ALS5* or *HWP1* genes compared to co-culture with the WT strain (except the expression of *ALS1* determined for 10C1B1I1 strain after 8 h of co-culture with NHDF). Opposite to NHDF, VK2 E6/E7 8 h co-culture led to a decrease in *ALS5* and *HWP1* gene expression in both KS058 and 10C1B1I1 strains compared to the WT strain (Figure 5C). The expression of *ALS1* and *HWP1* genes strongly increased for the 10C1B1I1 strain in prolonged co-cultures, which indicates that 24 h co-culture results in upregulating the expression of key adhesins genes (Figure 5D). For the KS058 strain, we observed elevated levels of *ALS5* expression. For the *ALS1* and *HWP1* genes, changes in expression were not statistically significant for KS058.

NHDF and VK2 E6/E7 cell lines differ in cytokines/chemokines production pattern. IL-1α is highly produced by VK2 E6/E7 cell line, especially after stimulation with *C. albicans* after both 8 and 24 h of co-culture. For the NHDF cell line, we did not detect the presence of IL-1α after 8 h. Only trace amounts of the cytokine were present after 24 h of co-culture with the *C. albicans* WT or 10C1B1I1 strains. *C. albicans* 10C1B1I1 triggers the significantly higher production of IL-6, IL-8 and MCP-1 compared to the WT strain. Interestingly, the NHDF cell line produced high amounts of MCP-1 in the control conditions, and even higher levels were detected after co-culture with investigated *C. albicans* strains (Table 2). This proves that fibroblasts produce high levels of this chemokine, and MCP-1 production is stimulated by fungal infections. The presence of high amounts of IL-8 is critical for the immune response of the host. The IL-8 is a strong chemoattractant and activator of neutrophils which are involved in combating fungal infections [43]. On the other hand, *C. albicans* KS058 infection resulted in a weak inflammatory response in the NHDF cell line after 8 and 24 h of co-culture, showing that morphological changes caused by depletion of ergosterol (e.g., reduced growth rate, impaired filamentation and biofilm formation) correspond with the weaker inflammatory response of the human cell lines.

In contrast to NHDF, VK2 E6/E7 cell line produced high amounts of IL-1α after contact with the WT or 10C1B1I1 strains (Table 2). This proves that fibroblasts and epithelial cell lines differ in their inflammatory response patterns, and co-culture with *C. albicans* mutants results in altered immune reactions. In epithelial cells, IL-1α is expressed constitutively, and elevated expression of interleukin is detected in response to inflammation [44]. Moreover, IL-1α also mediates the transcription of other cytokines e. g., IL-6 and IL-8 [45]. After 8 h of co-culture with VK2 E6/E7, both the WT and 10C1B1I1 strains stimulated cells for the production of similar levels of IL-6, IL-8 and MCP-1, while co-culture with the KS058 strain did not trigger the production of cytokines. The 24 h co-culture of VK2 E6/E7 with *C. albicans* 10C1B1I1 resulted in the detection of significantly higher levels of IL-1α than the WT strain. A similar tendency was observed in the case of IL-6 and IL-8 levels, which was confirmed by the increased expression of genes encoding those cytokines (Figure 5C,D). Interestingly, after 24 h of culture *C. albicans* KS058 triggered the inflammatory response of VK2 E6/E7, which produced 59.18, 6.64 and 24.71 pg/mL of IL-1α, IL-6 and IL-8, respectively (Table 2). This suggests that the induction of cytokines/chemokines production by *C. albicans* KS058 is delayed compared to the 10C1B1I1 or WT strains.

## 4. Materials and Methods

### 4.1. Reagents

During this study following reagents were used: yeast extract (YE), peptone (Becton Dickinson, Franklin Lakes, NJ, USA), glucose (Bioshop, Burlington, ON, Canada), phosphate-buffered saline (PBS) tablets, hexadecane, penicillin with streptomycin, L-Glutamine (L-Glu) (Merck, Darmstadt, Germany); RPMI 1640, KSFM medium with dedicated supplements (human recombinant EGF (rEGF), bovine pituitary extract (BPE)), fetal bovine serum (FBS) (Gibco, Thermo Fischer, Waltham, MA, USA); α-MEM medium (BioWest, Nuaillé, France); K_2_HPO_4_, KH_2_PO_4_ (Avantor Performance Materials, Gliwice, Poland), calcium chloride (CaCl_2_) (AppliChem GmbH, Darmstadt, Germany).

### 4.2. Strains, Cell Lines and Culture Condition

The *C. albicans* strains used in this study are CAF2-1 (genotype: *ura3Δ::imm434*/*URA3*), CAF4-2 (genotype: *ura3Δ::imm434*/*ura3Δ::imm434*) and SC5314 which were a kind gift from Prof. D. Sanglard (Lausanne, Switzerland) [46]. *C. albicans* KS058 was constructed by our team (the same genotype as SC5314 but *erg11Δ::SAT1-FLIP*/*erg11Δ::FRT*) [47] and 10C1B1I1 (the same genotype as SC5314 but *ERG11K143R::FRT*/*ERG11K143R::FRT*) which was a kind gift from Prof. D. Rogers (Department of Clinical Pharmacy, University of Tennessee Health Science Center, Memphis, TN, USA) [48]. Strains were routinely pre-grown at 28 °C, with shaking (120 rpm) for 24 h in a YPD medium containing 1% YE, 1% peptone and 2% glucose.

Cell lines NHDF (Lonza, Basel, Switzerland) and VK2 E6/E7 (ATCC, distributor: LGC, London, UK) were cultured according to distributor directions. NHDF was routinely cultured in complete α-MEM medium (10% FBS, 1% PenStrep, 1% L-Glu) in a humidified incubator (5% CO_2_, 37 °C). For the purpose of inflammatory response studies, NHDF were seeded into 24-well culture plates (Greiner Bio-One, Kremsmünster, Austria) in α-MEM with 1% FBS, 1% L-Glu and without antibiotics. During *C. albicans* co-cultures, NHDF were cultured in α-MEM with 1% L-Glu. VK2 E6/E7 were routinely cultured in complete KSFM medium supplemented with human rEGF (0.1 ng/mL), BPE (0.05 mg/mL), CaCl_2_ (0.4 mM) and 1% PenStrep in a humidified incubator (5% CO_2_, 37 °C). For the purpose of inflammatory response studies, VK2 E6/E7 was seeded into 24-well culture plates in a complete KSFM medium without antibiotics.

### 4.3. Cell Surface Hydrophobicity (CSH)

Cell surface hydrophobicity (CSH) was performed according to Biniarz et al. with modifications [39]. Twenty-four h cultures of *C. albicans* mutants (YPD, 5 mL, 28 °C, 120 rpm) were washed two times (4500 rpm; 5 min) and resuspended in phosphate buffer (16.9 g/L K_2_HPO_4_, 7.3 g/L KH_2_PO_4_). Cultures were adjusted to OD_600_ = 0.5 (±0.05) in 2 mL of phosphate buffer, and 500 μL of hexadecane was added. The probes were vigorously vortexed for 3 min and then incubated stationary for 60 min at RT. After this, OD_600_ of 1 mL of water phase was measured with Hach Odyssey DR/2500 spectrophotometer. The cell surface hydrophobicity (CSH) was calculated according to the formula (mean of 3 biological replicates):$$\mathrm{CSH}\;\left(\%\right)=100\times \left(1-\frac{\mathrm{OD}600\;t=60}{\mathrm{OD}600\;t=0}\right)$$

### 4.4. C. albicans Filamentation Assay

For the purpose of filamentation assay, *C. albicans* was cultured for 24 h in 20 mL of YPD medium supplemented with 20% FBS (starting OD_600_= 0.1). At different time points (at 0, 1, 2, 4, 6, 8 and 24 h of culture), *C. albicans* cells were harvested, washed 2 times with PBS (7500 rpm, 5 min) and concentrated. Then, microscopic preparations were observed under Zeiss Axio Imager (Oberkochen, Germany) A2 microscope. For the determination of hyphae length and the percentage of hyphae, at least *n* = 300 of *C. albicans* cells were analyzed in each condition.

### 4.5. Biofilm Formation Assay

The method was performed according to Lohse M. B. et al.’s protocol with modifications [49]. Briefly, *C. albicans* strains were pre-cultured in YPD for 24 h (28 °C, 120 rpm). Then cultures were washed 3-times with RPMI 1640 medium, pH 7.4 (4500 rpm, 5 min) and adjusted to OD_600_ = 1. 200 μL of prepared inoculum were transferred into a 96-well plate (Sarstedt, Nümbrecht, Germany) and then incubated for 90 min at 37 °C with agitation (250 rpm) using Spark microplate reader and incubator (Tecan, Männedorf, Switzerland). RPMI 1640 medium alone constitutes a negative control of the experiment. After initial incubation (adhesion of *C. albicans* cells to plate surface), the medium was discarded, wells were washed with PBS, and fresh RPMI 1640 medium was added. Then cells were cultured for 24 h under the same conditions (37 °C, 250 rpm).

The biofilm formation was determined by OD_600_ measurements of cultures: before incubation (t = 0 h), after initial incubation (t = 90 min) and after final incubation (t = 24 h; in order to remove non-adherent cells, the medium was discarded before reading) using Spark microplate reader (OD_600_ measurements were performed in 9 different points of the well and final OD_600_ value constitutes the average of measurements; the experiment was performed in 3 biological replicates).

### 4.6. Determination of Inflammatory Response of Human Cell Lines in Response to Infection with C. albicans Mutants

In order to determine the inflammatory response of used cell lines, the NHDF or VK2 E6/E7 were seeded into 24-well plates (Greiner Bio-One, Kremsmünster, Austria) at density 1 × 10^6^ cells per well in 500 μL of dedicated medium (Section 4.2) and cultured for 24 h in a humidified incubator (37 °C, 5% CO_2_). After this, the medium was discarded, and cells were inoculated with *C. albicans* mutants at the multiplicity of infection (MOI) = 10 (1 × 10^7^
*C. albicans* cells per well). Negative controls were human cells cultured alone in a dedicated medium.

After 8 or 24 h of co-culture, the medium was harvested and centrifuged (1500 rpm, 8 min) to remove non-adherent cells from the culture supernatant. After this, supernatants were tested for the presence of cytokines (IL-1α, IL-6, IL-8, IL-10 and MCP1) using ELISA–Milliplex MAP Kit (panel HCYTA-60; Merck, Darmstadt, Germany). The experiment was performed according to manufacturer instructions in 3 independent biological replicates. For each cytokine in the experiment, a standard curve was determined and R^2^ = 1.

### 4.7. Real-Time qPCR (RT-qPCR) Analysis of Adhesins and Cytokines Genes Expression

After co-culture of *C. albicans* strains with human cell lines, adherent cells were harvested by centrifugation (7500 rpm, 5 min). After this, cells were immediately resuspended in fenosol (A&A Biotechnology, Gdansk, Poland) and kept at –20 °C until the RNA isolation. The RNA isolation was performed using a Total RNA Mini kit (A&A Biotechnology, Gdansk, Poland) according to manufacturer instructions. Then RNA was adjusted to an equal concentration of 10 ng/μL, and reversed transcription (RT) was performed using a High-Capacity cDNA Reverse Transcription kit (Thermo Fisher Scientific, Waltham, USA). In order to determine the level of expression of genes encoding adhesins or cytokines, the cDNA was used as the matrix in real-time qPCR (RT-qPCR) and iTaq Universal SYBR Green Supermix (Bio-Rad, Hercules, CA, USA). RT-qPCR reaction was performed using Step-One Plus Real-Time PCR System (Applied Biosystems, Waltham, MA, USA). The list of used primers’ sequences is presented in Table 3.

For determination of adhesins genes expression (*ALS1*, *ALS5* and *HWP1*; reference gene: *ACT1*) following thermal cycling conditions were used: the initial step at 95 °C for 10 min, followed by 40 cycles at 95 °C for 20 s, 54 °C for 20 s and 72 °C for 30 s. For the determination of cytokine genes expression (*IL-1α*, *IL-6* and *IL-8*; reference gene: *GAPDH*), RT-qPCR was performed as follows: the initial step at 95 °C for 10 min, followed by 40 cycles at 95 °C for 20 s, 59 °C for 20 s and 72 °C for 30 s. The relative expression of target genes was calculated using the 2^−*∆∆*Ct^ method.

### 4.8. Statistical Analysis

Unless stated otherwise, data represent the means ± standard errors from at least 3 biological replicates. Statistical significance was determined using Student’s *t*-test (binomial, unpaired).

## Figures and Tables

**Figure 1 ijms-24-03966-f001:**
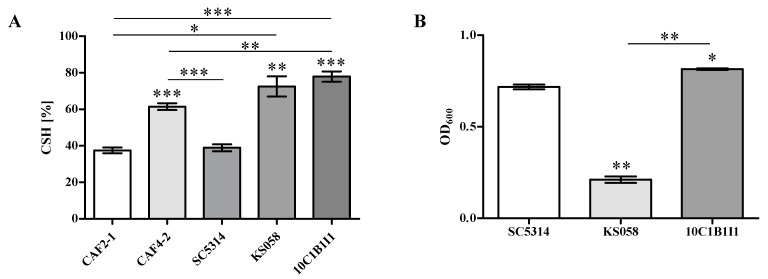
The cell surface hydrophobicity (CSH) and biofilm formation of *C. albicans* strains. (**A**) CSH (%) of *C. albicans* strains: CAF2-1 (genotype: *ura3Δ::imm434*/*URA3*), CAF4-2 (genotype: *ura3Δ::imm434*/*ura3Δ::imm434*; parental strain: CAF2-1), SC5314, KS058 (*erg11Δ*/*Δ*, parental strain: SC5314) and 10C1B1I1 (*ERG11^K143R^*^/*K143R*^; parental strain: SC5314) after 24 h of culture (YPD, 28 °C, 120 rpm); *n* = 3, ±SD; in each case the strains were compared to its parental strain as follows: CAF4-2 to CAF2-1; KS058 and 10C1B1I1 to SC5314 or to each other (* *p* < 0.05; ** *p* < 0.01; *** *p* < 0.001). (**B**) Biofilm formation of *C. albicans* WT (SC5314), KS058 and 10C1B1I1 strains expressed as OD_600_ after 24 h incubation (*n* = 3, ±SD); data were compared to WT strain or to each other (* *p* < 0.05; ** *p* < 0.01).

**Figure 2 ijms-24-03966-f002:**
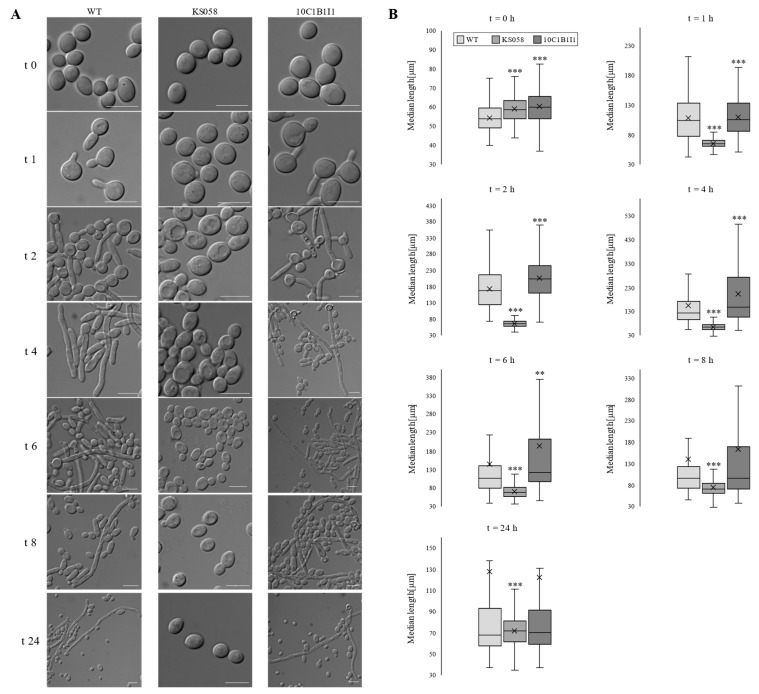
Filamentation of *C. albicans* WT (SC5314), KS058 (*erg11Δ*/*Δ*) and 10C1B1I1 (*ERG11^K143R^*^/*K143R*^) strains in the presence of 20% fetal bovine serum (FBS) in YPD medium. (**A**) Microscopic study of *C. albicans* strains performed in different time points of culture in the presence of 20% FBS (time points: 0, 1, 2, 4, 6, 8 and 24 h of culture). The figure presents representative micrographs for each condition (scale bar 100 µm). (**B**) Median length of *C. albicans* WT, KS058 and 10C1B1I1 cells measured in different time points of culture with 20% FBS (*n* = 300). The “x” on the median bars represents mean values. Mean data (±SD) collected for *C. albicans* KS058 and 10C1B1I1 were compared to those obtained for the WT strain (** *p* < 0.01; *** *p* < 0.001).

**Figure 3 ijms-24-03966-f003:**
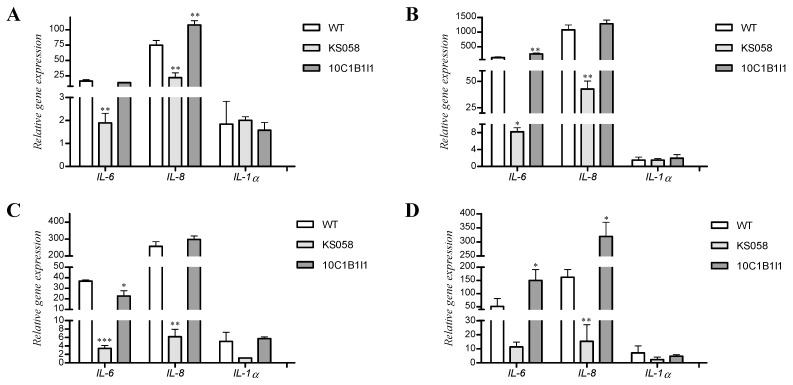
Relative expression of genes encoding cytokines (*IL-6, IL-8* and *IL-1α*) determined for NHDF after 8 (**A**) or 24 h (**B**) and for VK2 E6/E7 cell lines after 8 (**C**) or 24 h (**D**) of co-culture with *C. albicans* WT (SC5314), KS058 (*erg11Δ*/*Δ*) and 10C1B1I1 (*ERG11^K143R^*^/*K143R*^) strains (*n* = 3; ±SD). Results determined for KS058 or 10C1B1I1 strains were compared to the WT strain for certain cytokine genes (* *p* < 0.05; ** *p* < 0.01; *** *p* < 0.001).

**Figure 4 ijms-24-03966-f004:**
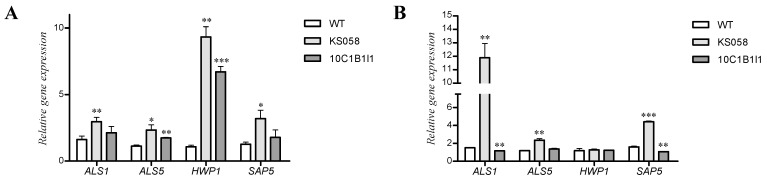
Relative expression of genes encoding adhesins (*ALS1, ALS5, HWP1, SAP5*) determined for *C. albicans* WT (SC5314), KS058 (*erg11Δ*/*Δ*) or 10C1B1I1 (*ERG11^K143R^*^/*K143R*^) strains after 8 h (**A**) or 24 h (**B**) of culture in YPD medium (*n* = 3; ±SD). Results obtained for KS058 or 10C1B1I1 strains were compared to the WT strain for certain adhesin genes (* *p* < 0.05; ** *p* < 0.01; *** *p* < 0.001).

**Figure 5 ijms-24-03966-f005:**
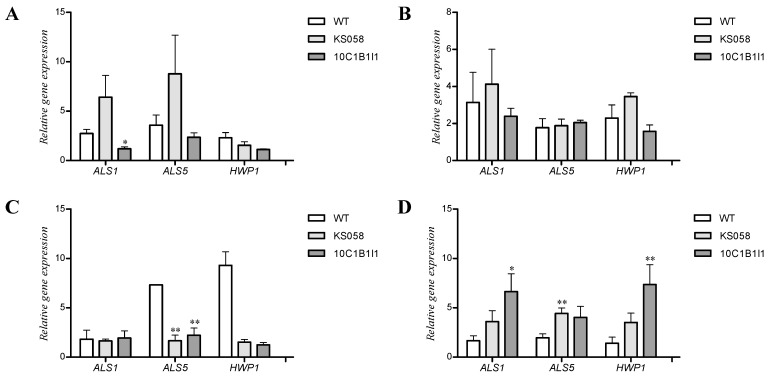
Relative expression of genes encoding adhesins (*ALS1*, *ALS5*, *HWP1*) determined for *C. albicans* WT (SC5314), KS058 (*erg11Δ*/*Δ*) or 10C1B1I1 (*ERG11^K143R^*^/*K143R*^) strains for NHDF after 8 (**A**) or 24 h (**B**) and for VK2 E6/E7 cell lines after 8 (**C**) or 24 h (**D**) of co-culture (*n* = 3; ±SD). Results obtained for KS058 or 10C1B1I1 strains were compared to the WT strain for certain cytokine genes (* *p* < 0.05; ** *p* < 0.01).

**Table 1 ijms-24-03966-t001:** Percent of hyphae cells (%) in *C. albicans* WT (SC5314) and 10C1B1I1 (*ERG11^K143R^*^/*K143R*^) strains culture with the addition of FBS (20%) in different time points (0, 1, 2, 4, 6, 8 and 24h). For this purpose, micrographs were analyzed for mean percent of hyphae (*n* = 300; ±SD); Results obtained for *C. albicans* 10C1B1I1 strain in different time points were compared to WT strain (* *p* < 0.05; ** *p* < 0.01).

Time of Culture (h)	Percent of Hyphae (%)	
	WT	10C1B1I1
0	0.00 ± 0.00	0.00 ± 0.00
1	62.73 ± 4.65	52.29 ± 5.81
2	64.82 ± 3.01	72.29 ± 3.18 *
4	56.68 ± 1.81	51.73 ± 1.58 *
6	36.70 ± 3.19	27.04 ± 0.47 *
8	25.61 ± 3.07	25.22 ± 0.94
24	9.18 ± 1.01	5.42 ± 0.64 **

**Table 2 ijms-24-03966-t002:** Cytokines and chemokines levels (IL-1α, IL-6, IL-8, IL-10, MCP-1) detected in culture supernatants released in response to infection with *C. albicans* WT (SC5314), KS058 (*erg11Δ*/*Δ*) and 10C1B1I1 (*ERG11^K143R^*^/*K143R*^) strains after 8 or 24 h of co-culture with NHDF or VK2 E6/E7 cell lines (*n* = 3; ±SD). Results obtained for WT, KS058 and 10C1B1I1 strains was compared to control conditions of NHDF or VK2 E6/E7 cell lines cultured alone among different time of culture (* *p* < 0.05; ** *p* < 0.01; *** *p* < 0.001).

Cell Line	Time of co-culture (h)	*C. albicans* Strain	IL-1α [pg/mL]	IL-6 [pg/mL]	IL-8 [pg/mL]	IL-10 [pg/mL]	MCP-1 [pg/mL]
NHDF	8	Control	ND	1.48 ± 0.74	0.15 ± 0.09	0.37 ± 0.28	4.50 ± 2.58
		WT	ND	20.45 ± 1.96 ***	3.16 ± 0.23 ***	0.21 ± 0.21	13.98 ± 0.78 ***
		KS058	ND	2.60 ± 0.32 *	0.39 ± 0.05 **	0.01 ±0.02	6.00 ± 0.77
		10C1B1I1	ND	24.14 ± 1.65 ***	8.69 ± 0.35 ***	0.40 ± 0.24	18.04 ± 0.55 ***
	24	Control	ND	10.55 ± 3.8	0.61 ± 0.12	0.30 ± 0.20	277.38 ± 96.11
		WT	1.83 ± 0.11 ***	214.83 ± 64.57 **	39.52 ± 10.67 **	0.19 ± 0.26	476.25 ± 24.30 *
		KS058	ND	14.19 ± 6.76	3.26 ± 1.28 *	0.25 ± 0.11	314.52 ± 16.90
		10C1B1I1	2.88 ± 0.42 ***	242.84 ± 88.39 *	52.34 ± 13.96 **	0.58 ± 0.12	405.78 ± 20.10
VK2 E6/E7	8	Control	5.80 ± 0.65	0.69 ± 0.11	2.60 ± 0.60	0.13 ± 0.23	5.24 ± 0.13
		WT	1612.60 ± 442.55 **	13.13 ± 0.74 ***	31.56 ± 6.00 ***	0.74 ± 0.32 *	3.24 ± 0.35 ***
		KS058	5.66 ± 0.89	1.38 ± 0.36 *	3.78 ± 0.04 *	0.01 ± 0.02	4.90 ± 0.16 **
		10C1B1I1	1530.79 ± 529.76 *	16.66 ± 1.61 ***	39.60 ± 6.98 ***	0.37 ± 0.22	2.60 ± 0.72 **
	24	Control	12.57 ± 2.79	1.47 ± 0.42	5.37 ± 0.79	0.15 ± 0.22	3.56 ± 0.43
		WT	1713.36 ± 793.24	12.26 ± 0.25 ***	205.47 ± 22.74 ***	0.36 ± 0.14	2.62 ± 0.19 **
		KS058	59.18 ± 29.43 *	6.64 ± 0.68 ***	24.71 ± 9.99 *	0.16 ± 0.21	3.31 ± 1.03
		10C1B1I1	2476.71 ± 863.84 *	15.10 ± 1.12 **	230.04 ± 27.49 ***	0.42 ± 0.31	1.60 ± 0.18 ***

**Table 3 ijms-24-03966-t003:** Sequences of primers used for RT-qPCR in this study.

Primer Name	Sequence (5′ → 3′)
ACT1_F	TCC AGC TTT CTA CGT TTC CA
ACT1_R	GTC AAG TCT CTA CCA GCC AA
ALS1_F	TGT TGG TGT GAC TAC TTC CT
ALS1_R	TGT ACC ACC ACT GTG TCA AT
ALS5_F	GTT CAG ACA TGC CAT CAT CG
ALS5_R	CTC CAA GTG ATC AGA GTG GA
HWP1_F	ACC ACT ACT ACT GAA GCC AAA
HWP1_R	CTG GAG CAG TAG AAA CTG GA
GAPDH_F	TGA ACG GGA AGC TCA CTG G
GAPDH_R	TCC ACC ACC CTG TTG CTG TA
IL-1α_F	GTC TCA CTT GTC TCA CTT GTG
IL-1α_R	GGT AGC CAT AGT CAG TAG CTC
IL6_F	AAA GAG GCA CTG GCA GAA AA
IL6_R	TTT CAC CAG GCA AGT CTC CT
IL8_F	TGG CTC TCT TGG CAG CCT TC
IL8_R	TGC ACC CAG TTT TCC TTG GG

## Data Availability

Data sharing not applicable. No new data were created or analyzed in this study. Data sharing is not applicable to this article.

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
