# Peer review of "Alterations in the Level of Ergosterol in Candida albicans’ Plasma Membrane Correspond with Changes in Virulence and Result in Triggering Diversed Inflammatory Response"

_ijms, 2023, doi:10.3390/ijms24043966_

Round 1

Reviewer 1 Report

In the manuscript ‘Alterations in the level of ergosterol in Candida albicans’ plasma membrane correspond with changes in virulence and result in triggering diversed inflammatory response’, the authors performed the set of experiments, proving that C. albicans Erg11 mutants, resistant to the antifungal drugs, azoles, are affected in the key virulence factors, which reflects inflammatory response of the host cells. Both, C. albicans deletion erg11∆/∆ and ERG11K143R/K143R mutant, have increased cell surface hydrophobicity, and the former is truncated also in biofilm and hyphae formation. Furthermore, both mutants trigger opposed inflammatory response of human cell lines, where the most pronounced effect was observed in a case of point mutant. Moreover, expressions of the genes encoding adhesins were also altered in such mutants.

In my opinion, presented work is interesting and it is important to the other fungal researchers, working on Candida species. The carefully thought-out experiments gave the interesting results, although I see few minor points concerning this work, which makes it unclear, as listed below:

Row 85: K143R mutation refers to protein, not the ERG11 gene

Figure 1: what is genetic background, which differentiates both CAF strains? It should  be explained

Row 207-8: please, re-write this sentence. It is unclear to understand

Row 246: this sentence is too long. Please, split it

Row 258: to re-write: suppose that the higher value refers to expression, not Candida nor stimulation

Row 397-8: this sentence contradicts the results presented in the figure 5. Moreover, not discussed with the literature

Row 448: where are CAF strains? The genotype and the source should be mentioned

Row 466: the reference of this method should be mentioned

Row 500 and 502: it is inflammatory response rather than immune

Small errors: row 19-results, row 57-CW is the first (barrier?), row 74: to combat (against?), row 82-encoded by the ERG11 gene, row 85-alternation in ERG11 expression, row 106-in the virulence, row 143-[6], row 262-expression pattern of...,row 328-it is first one to

Author Response

In the manuscript ‘Alterations in the level of ergosterol in Candida albicans’ plasma membrane correspond with changes in virulence and result in triggering diversed inflammatory response’, the authors performed the set of experiments, proving that C. albicans Erg11 mutants, resistant to the antifungal drugs, azoles, are affected in the key virulence factors, which reflects inflammatory response of the host cells. Both, C. albicans deletion erg11∆/∆ and ERG11K143R/K143R mutant, have increased cell surface hydrophobicity, and the former is truncated also in biofilm and hyphae formation. Furthermore, both mutants trigger opposed inflammatory response of human cell lines, where the most pronounced effect was observed in a case of point mutant. Moreover, expressions of the genes encoding adhesins were also altered in such mutants.

In my opinion, presented work is interesting and it is important to the other fungal researchers, working on Candida species. The carefully thought-out experiments gave the interesting results, although I see few minor points concerning this work, which makes it unclear, as listed below:

Row 85: K143R mutation refers to protein, not the ERG11 gene

Thank you for that remark. We deleted the fragment “one of the most frequent is K143R amino acid substitution” in row 84-5 to avoid the misunderstanding.

Figure 1: what is genetic background, which differentiates both CAF strains? It should be explained. Row 448: where are CAF strains? The genotype and the source should be mentioned

Thank you for pointing that out. The genetic background of C. albicans CAF2-1 and CAF4-2 is ura3Δ::imm434/URA3 and ura3Δ::imm434/ura3Δ::imm434, respectively.

In section 4.2. (row: 440-442) we added the detailed information about source and genetic background of C. albicans CAF2-1 and CAF4-2 strains. We also added the genotype of CAF2-1 and CAF4-2 strains in the description of Figure 1 (lines: 115-116).

Row 207-8: please, re-write this sentence. It is unclear to understand.

At row 204-5 we rewrote the sentence according to your suggestion. We changed it for: “The higher concentration of IL-8 was observed in case of NHDF co-cultured with C. albicans WT or 10C1B1I1 strains (3.16 or 8.69 pg/mL, respectively).”

Row 246: this sentence is too long. Please, split it.

In row 243-6 we split the long sentence in order to make it clearer. We changed it for: “The research indicates that stimulation of NHDF cell line with C. albicans WT or 10C1B1I1 strains for 8 hours results in increased expression of IL-6 and IL-8 genes (Figure 3A). At the same time, the higher expression of IL-8 gene was observed for co-culture with C. albicans 10C1B1I1 comparing to WT strain.”

Row 258: to re-write: suppose that the higher value refers to expression, not Candida nor stimulation

Thank you for the suggestion. We rewrote that sentence in order to clarify the statement. We changed it for: “The expression of IL-8 gene in VK2 E6/E7 cells after 24 hours of stimulation with C. albicans 10C1B1I1 was significantly higher than after stimulation with WT strain (Figure 3D). After 8 hours of co-culture with WT or 10C1B1I1 strains we observed a similar level of IL-8 expression (Figure 3C).” (row: 255-8).

Row 397-8: this sentence contradicts the results presented in the figure 5. Moreover, not discussed with the literature

Thank you for pointing this out. Here, we meant to emphasize that there are no significant changes in expression of genes encoding adhesins between investigated C. albicans strains after NHDF co-culture. We decided to delete the confusing part and express our observations in the sentence: “In case of C. albicans KS058 or 10C1B1I1 co-culture with NHDF we did not observe significant changes in expression of ALS1, ALS5 or HWP1 genes comparing to co-culture with WT strain (except the expression of ALS1 determined for 10C1B1I1 strain after 8 hours of co-culture with NHDF)” (lines: 387-390).

Row 466: the reference of this method should be mentioned

Thank you for this remark. We added the proper reference for used method in sentence: “Cell surface hydrophobicity (CSH) was performed according to Biniarz et al. with modifications [39].” (row 461-2).

Row 500 and 502: it is inflammatory response rather than immune

Thank you for pointing that out. We changed “immune response” for “inflammatory response” at the section title: “Determination of inflammatory response of human cell lines in response to infection with C. albicans mutants” (row 496-7) and in the sentence: “In order to determine the inflammatory response of used cell lines…” (row 498).

Small errors: row 19-results, row 57-CW is the first (barrier?), row 74: to combat (against?), row 82-encoded by the ERG11 gene, row 85-alternation in ERG11 expression, row 106-in the virulence, row 143-[6], row 262-expression pattern of...,row 328-it is first one to

We corrected mentioned small error. This including changes:

from “result” to “results” (row 19),

from „CW is the first to interfere..” to „CW is the first barrier to interfere..” (row 57),

from „effective way to combat C. albicans infections” to „effective way to combat against C. albicans infections” (row 74),

from „encoded by ERG11 gene..” to „encoded by the ERG11 gene..” (row 82),

from „manipulation in ERG11 expression” to „alternation in ERG11 expression” (row 85),

from “in virulence” to “in the virulence” (row 102),

from “[6i]” to “[6]” (row 140),

from “pattern of expression of” to “expression pattern of...” (row 259),

from “it is first to interfere..” to „it is first one to interfere..” (row 325).

Reviewer 2 Report

11.     Authors should add details about Institutional Review Board Statement.

22.     Investigators used two mutant strains - KS058 and 10C1B1I1 in the study. The characteristics of these strains should be included in the introduction part.

33.     Investigators observed that both mutant strains - KS058 and 10C1B1I1 exhibit significantly elevated cell surface hydrophobicity (72.50 and 77.95%, respectively). But KS058 has low ergosterol content and 10C1B1I1 has high ergosterol content in the PM. Then what could be the possible reason for high CSH of KS058 strain? It may be added in the discussion part.

44.     It is requested to add reference for the cell surface hydrophobicity (CSH) method used in the study.

55.    Authors have described results of the study in the last paragraph of the introduction part (Page no. 2 line no. 92 to page no. 3 line no. 109). It is suggested to revise the paragraph. Results may be shifted to either result part or discussion part. Authors may describe the purpose or objectives of various experiments in the introduction part but should be without results.

66.     Figure 1 (Page no. 3 and line no. 118) is in italics but Figure 4 (Page no. 9 and line no. 272) is not. Authors are advised to follow journals recommendation and maintain uniformity and modify similar changes in other legends.

77.     C. albicans (Page no. 3 and line no. 118) should be in italics. Authors are advised to correct similar changes in other legends.

88.     The information on page no. 5 from line no. 161 to 166 is repeated on the same page from line no. 173 to 176.

99     It is requested to recheck the p value of cytokines and chemokines levels after 8 hours with VK2 E6/E7 cell lines depicted in table 2.

110.  It is requested to recheck the p value of relative expression of genes encoding cytokines levels after 24 hours with NHDF cell lines depicted in figure 3 (B).

111.  It is requested to recheck the p value of relative expression of genes encoding adhesins levels after 8 hours depicted in figure 4 (A).

112.  Page no. 10 and line no. 325 to 336 is the repetition of results. Similarly, Page no. 11 and line no. 358 to 369 is the prepetition of results. Authors are advised to rewrite it.

113.  The statement on page number 12 and line number 404-405 – “For KS058 strain we observe elevated level of ALS1, ALS5 and HWP1 but those changes are not statistically significant” is not matching with figure 5 (D).

114.  The statement on page number 12 and line number 407-409 – “In cell culture supernatant we did not detect the IL-1α presence in none of the tested conditions. On the other hand, we observed the similar trend in level of IL-6, IL-8 and MCP-1 after co-culture with C. albicans mutants.” is not matching with table 2.

115.  References should be cited as per journal’s recommendation - issue number may not be needed.

Author Response

  1. Authors should add details about Institutional Review Board Statement.

Thank you for that comment. We added the information about Institutional Review Board Statement in line 541.

  1. Investigators used two mutant strains - KS058 and 10C1B1I1 in the study. The characteristics of these strains should be included in the introduction part.

Thank you for that comment. The characteristic of C. albicans KS058 and 10C1B1I1 strains is now more specified in the sentence “We previously reported that both point mutation (K143R) in ERG11 gene (in C. albicans 10C1B1I1 strain) or deletion of this gene (in C. albicans KS058 strain) corresponds with changes in plasma membrane (PM) and CW composition of C. albicans mutants” (lines 86-8). The genotype of strains used in this study is also described in lines 91-2.

  1. Investigators observed that both mutant strains - KS058 and 10C1B1I1 exhibit significantly elevated cell surface hydrophobicity (72.50 and 77.95%, respectively). But KS058 has low ergosterol content and 10C1B1I1 has high ergosterol content in the PM. Then what could be the possible reason for high CSH of KS058 strain? It may be added in the discussion part.

Thank you for this question. We previously reported that both KS058 and 10C1B1I1 strains display an altered cell wall structure by exposure of β-glucan and chitin on the cell surface [1]. The CSH has been linked to the character of the cell wall architecture [2]. Therefore, the alteration in C. albicans mutants cell wall (different composition of cell wall components and potential alteration in the cell wall protein architecture) can impact the CSH of investigated strains. We added that conclusion in the discussion part according to your suggestion (lines 330-335).

  1. It is requested to add reference for the cell surface hydrophobicity (CSH) method used in the study.

Thank you for this remark. We added the proper reference for used CSH method in section 4.3 in the sentence: “Cell surface hydrophobicity (CSH) was performed according to Biniarz et al. with modifications [39].” (lines: 461-462)

  1. Authors have described results of the study in the last paragraph of the introduction part (Page no. 2 line no. 92 to page no. 3 line no. 109). It is suggested to revise the paragraph. Results may be shifted to either result part or discussion part. Authors may describe the purpose or objectives of various experiments in the introduction part but should be without results.

Thank you for this comment. We deleted the description of the results from the last paragraph of introduction part. We rewrote it in order to describe only the purpose of performed experiments according to your suggestions (lines 91-104).

  1. Figure 1 (Page no. 3 and line no. 118) is in italics but Figure 4 (Page no. 9 and line no. 272) is not. Authors are advised to follow journals recommendation and maintain uniformity and modify similar changes in other legends.

Thank you for this comment. According to Your suggestions we unified the style of legends in Figure 1 (line: 114), Figure 2 (line: 146), Table 1 (line: 161), Table 2 (line: 182), Figure 3 (line: 238), Figure 5 (line: 287) and Table 3 (line: 525).

  1.    C. albicans (Page no. 3 and line no. 118) should be in italics. Authors are advised to correct similar changes in other legends.

We apologize for our mistake. We already applied italics to all legends (lines: 114-121, 146-153, 161-165, 182-187, 238-242, 269-271 and 287-291) in case of description of C. albicans strains, according to you suggestion.

  1.    The information on page no. 5 from line no. 161 to 166 is repeated on the same page from line no. 173 to 176.

Thank you for the remark. We decided to delete the repeated sentence in lines 161-166 and move that statement to lines 168-172.

  1. It is requested to recheck the p value of cytokines and chemokines levels after 8 hours with VK2 E6/E7 cell lines depicted in table 2.

We rechecked the p values determined for cytokines and chemokines levels after 8 hours with VK2 E6/E7 cell lines. We added the asterisks in case of IL-10 determined for VK2 E6/E7 cell line co-culture with WT strain (p value = 0.013) in the Table 2. Other p values are correct.

  1. It is requested to recheck the p value of relative expression of genes encoding cytokines levels after 24 hours with NHDF cell lines depicted in figure 3 (B).

We rechecked the p values determined for relative expression of genes encoding cytokines levels after 24 hours with NHDF cell line (Figure 3B) and it is correct.

  1. It is requested to recheck the p value of relative expression of genes encoding adhesins levels after 8 hours depicted in figure 4 (A).

We rechecked the p values determined for relative expression of genes encoding adhesins levels after 8 hours in YPD media (Figure 4A) and it is correct.

  1. Page no. 10 and line no. 325 to 336 is the repetition of results. Similarly, Page no. 11 and line no. 358 to 369 is the prepetition of results. Authors are advised to rewrite it.

Thank you for this comment. In lines 325-336 we left only the essential conclusions from CSH experiment. The sentences are now: “The analysis revealed that comparing to C. albicans WT (SC5314) both KS058 and 10C1B1I1 strains demonstrate significantly higher CSH and the CSH obtained for those mutants is even higher than for hydrophobic CAF4-2 strain (Figure 1A).” (lines: 326-328).

We also deleted the repeated fragment of results in lines 358-369 and left only the general conclusions from that experiment. The statement is now: “The filamentation assay revealed that for C. albicans WT and 10C1B1I1 strains the creation of filaments initiated after 1 hours of culture in YPD media supplemented with 20% FBS (Figure 2A). After 4 hours of culture, major number of C. albicans WT cells produce hyphae, while for C. albicans 10C1B1I1 strain we observed the mixed culture of blastoconidia and hyphae. Additionally, at early phase of culture the median hyphae length was significantly higher for C. albicans 10C1B1I1 comparing to WT strain (Figure 2B)” (lines: 355-358).

  1. The statement on page number 12 and line number 404-405 – “For KS058 strain we observe elevated level of ALS1, ALS5 and HWP1 but those changes are not statistically significant” is not matching with figure 5 (D).

Thank you for this comment. We rewrote this sentence to match results presented in Figure 5D, according to your suggestion. Now the statement is: “For KS058 strain we observe elevated level of ALS5 expression. For ALS1 and HWP1 genes those changes in expression are not statistically significant for KS058 strain.” (lines: 394-396).

  1. The statement on page number 12 and line number 407-409 – “In cell culture supernatant we did not detect the IL-1α presence in none of the tested conditions. On the other hand, we observed the similar trend in level of IL-6, IL-8 and MCP-1 after co-culture with C. albicansmutants.” is not matching with table 2.

Thank you for the comment. We change the sentences: “In cell culture supernatant we did not detect the IL-1α presence in none of the tested conditions. On the other hand, we observed the similar trend in level of IL-6, IL-8 and MCP-1 after co-culture with C. albicans mutants.” for “IL-1α is highly produced by VK2 E6/E7 cell line, especially after stimulation with C. albicans after both 8 and 24 hours of co-culture. For NHDF cell line we did not detect the IL-1α presence after 8 hours. Only the trace amount of the cytokine was present after 24 hours of co-culture with C. albicans WT or 10C1B1I1 strains” (lines: 398-401).

  1. References should be cited as per journal’s recommendation - issue number may not be needed.

Thank you for pointing that out. We modified the references according to journal’s recommendation.

[1] Derkacz D., Bernat P., Krasowska A. K143R Amino Acid Substitution in 14-α-Demethylase (Erg11p) Changes Plasma Membrane and Cell Wall Structure of Candida albicans. Int J Mol Sci. 2022, 23, 1631.

[2] Masuoka J., Hazen K. C. Cell wall mannan and cell surface hydrophobicity in Candida albicans serotype A and B strains. Infect Immun. 2004, 72, 6230-6.